# Estimation of the heritability of medicinal compound contents in *Glycyrrhiza uralensis*

**Takahiro Tsusaka**(ID)*, **Miki Sakurai**

Botanical Raw Materials Cultivation Technology Development Department, Tsumura & Co., Ami, Ibaraki, Japan

* tsusaka_takahiro@mail.tsumura.co.jp

## Abstract

*Glycyrrhiza uralensis* Fischer (Fabaceae), an important medicinal plant widely used in traditional Japanese and Chinese medicine, contains bioactive compounds, such as glycyrrhizin and flavonoids, which exhibit pharmacological activities, including antispasmodic and antitussive effects. Overharvesting has depleted wild populations, making the cultivation of *G. uralensis* necessary to stabilize its supply; however, the content of bioactive compounds tends to decrease in cultivated plants. In this study, we investigated the genetic inheritance (heritability) of medicinal compound contents in *G. uralensis* by analyzing clonal lines propagated from stolons. Broad-sense heritability was estimated for seven medicinal compounds using 26 clonal lines, revealing high heritability for glycyrrhizin and several flavonoids. In addition, correlation analyses between seed-derived and stolon-derived roots demonstrated strong genetic inheritance of these compounds. Furthermore, the effects of plant age and cultivation year on compound content were examined. Our results show high correlations between compound content and genotype across plant age (one- and two-year-old plants), suggesting the feasibility of early selection in breeding programs. While environmental variations influenced absolute compound levels, the relative rankings among genotypes remained stable. These results indicate the high heritability of compound contents and the strong effectiveness of selective breeding. We conclude that there is strong potential for selective breeding to enhance the medicinal quality of *G. uralensis* by targeting high-yielding genotypes with superior compound profiles and that this potential may also apply to the sustainable cultivation of other high-quality medicinal plant resources.

## Introduction

*Glycyrrhiza uralensis* is one of the most important medicinal plants used in Kampo (Japanese traditional medicine) and Chinese traditional medicine; the roots and stolons of the plant are used as botanical raw materials for many prescriptions in

**Data availability statement:** All relevant data are within the paper and its Supporting Information files.

**Funding:** The author(s) received no specific funding for this work.

**Competing interests:** The authors have declared that no competing interests exist.

traditional medicines. Moreover, *G. uralensis* is also utilized as an ingredient in food additives, cosmetic products, and flavor additives in tobacco and confectionery [1,2]. The primary bioactive compounds in the roots and stolons of *G. uralensis* are triterpenoids and flavonoids; in particular, glycyrrhizin, a specific triterpenoid, stands out for its significant pharmacological activity and its utility as a raw material in sweeteners [3]. Glycyrrhizin is known to exhibit a range of pharmacological effects, including anti-inflammatory, antioxidative, antiviral, and hepatoprotective properties [4–6]. Flavonoids, namely liquiritin, liquiritin apioside, isoliquiritin, isoliquiritin apioside, liquiritigenin, and isoliquiritigenin, are significant compounds in *G. uralensis* [1]. Each of these compounds has been reported to exhibit distinct pharmacological effects. Specifically, liquiritigenin, liquiritin, and liquiritin apioside are known for their antitussive properties, and isoliquiritigenin, isoliquiritin, and isoliquiritin apioside have been identified for their antispasmodic effects [7–9]. Liquiritigenin is the aglycone of liquiritin and liquiritin apioside, which are metabolized into liquiritigenin in the body. Similarly, isoliquiritigenin is the aglycone of isoliquiritin and isoliquiritin apioside, which are also hydrolyzed to isoliquiritigenin. These aglycones are responsible for the primary pharmacological activities in the body. Therefore, to obtain a reliable and stable level of biological activity, we must first carry out a comprehensive analysis of these compounds, including their glycosides [10].

*G. uralensis* is mainly distributed in China, Mongolia, Russia, and neighboring regions and is typically found growing in arid environments [1,11]. In recent years, overharvesting of wild populations has raised concerns about the overall depletion of *G. uralensis* as a valuable natural resource. In addition, *G. uralensis* plays an important role in preventing desertification by stabilizing sandy soil and maintaining soil moisture. As a result, the Chinese government has begun implementing export regulations to conserve this species and prevent desertification [11]. Although the cultivation of *G. uralensis* has been promoted, the content of medicinal constituents, particularly glycyrrhizin and liquiritin, tends to decrease in cultivated plants compared to their wild plants [2,12]. In both the Japanese Pharmacopoeia and the Pharmacopoeia of the People's Republic of China, the glycyrrhizin content in Glycyrrhizae Radix must be regulated at a minimum of 2.0% on a dry weight basis for medicinal use [13,14]. However, the glycyrrhizin content in cultivated plants sporadically falls below this threshold. Thus, it is necessary to increase the content of medicinal constituents, such as glycyrrhizin and liquiritin, in cultivated *G. uralensis* [15].

The production of medicinal constituents in *G. uralensis* has been studied to date, considering both environmental and genetic factors. As for the effects of environmental factors, it has been reported that glycyrrhizin content varies due to low-temperature stress, $Ca^{2+}$ concentrations in the soil, and the gaseous phase ratio of the soil, with low temperatures having a particularly significant impact on increasing glycyrrhizin contents [16,17]. Additionally, the construction of chromosome-scale genome sequences and the isolation of enzymes involved in glycyrrhizin biosynthesis have progressed, and both the biosynthetic pathways of glycyrrhizin and flavonoids, along with the genomic locations of associated genes, have been elucidated [18–20]. Expression analysis of the biosynthetic genes of glycyrrhizin and flavonoids has

revealed that the genes are induced by drought stress and the plant hormone jasmonic acid [21,22]. This suggests that the content of medicinal constituents in *G. uralensis* fluctuates due to environmental factors. In contrast, regarding genetic factors, it has been reported that the glycyrrhizin and liquiritin contents of five-year-old *G. uralensis* plants cultivated from seeds under the same conditions show large variations. This indicates that the effect of genetic factors on the medicinal constituents in *G. uralensis* is significant, resulting in large variations in glycyrrhizin and liquiritin contents [15]. In addition, while there have been attempts to create cultivars with high glycyrrhizin content [11], there are few extensive genetic analyses of *G. uralensis.*

*G. uralensis* is highly likely to exhibit inbreeding depression, making genetic fixation difficult [11]. Therefore, genetic analyses associated with the contents of medicinal constituents have not been conducted using genetically homogenized lines in *G. uralensis*. Alternatively, vegetative propagation through tissue culture and division of stolons in *G. uralensis* is possible, suggesting that genetic evaluation could be conducted using clonal lines [23,24].

In the present study, to evaluate the effects of genetic factors on the content of medicinal constituents in *G. uralensis* accurately, twenty-four clonal lines were established by division of stolons from 240 individual plants cultivated from seeds. The clonal lines were cultivated in a micro-environment, and the broad-sense heritability of the content of seven medicinal constituents, including glycyrrhizin and liquiritin, was estimated. To examine the genotype-environment (G × E) interaction and the effects of environmental factors, fourteen lines were cultivated in different years (2022, 2023), and the medicinal constituents were analyzed. In addition, to explore the potential for reducing the cultivation period required for evaluation in selective breeding of *G. uralensis*, a comparative analysis was conducted between first- and second-year plants. Lastly, a correlation analysis of the seven constituents in *G. uralensis* was performed to assess the potential for linkage during selection. Through these studies, the heritability of medicinal constituent contents in *G. uralensis* was evaluated.

## Materials and Methods

### Plant materials and cultivation

*G. uralensis* seeds obtained from the seed bank of Tsumura and Co., Ibaraki, Japan, were used. The seeds (Accession No. THS92532), originally from China, were introduced into Japan in 1987 and have since been preserved in the seed bank. The seeds were sown in the experimental field located in Ami-machi, Inashikigun, Ibaraki (35°.99'N, 140°.20'E), Japan, in 2020, and 240 plants were grown over a period of two years. The plants were harvested on November 21, 2021. The main roots of the plants were used to evaluate medicinal compound contents, while the stolons were utilized for the propagation of clonal lines (S1 Fig). A clonal line of *G. uralensis* was propagated from the stolons of a single plant; the 31 clonal lines were labeled as follows: Tm21_013, Tm21_016, Tm21_020, Tm21_026, Tm21_028, Tm21_029, Tm21_030, Tm21_035, Tm21_036, Tm21_037, Tm21_045, Tm21_049, Tm21_051, Tm21_052, Tm21_055, Tm21_058, Tm21_060, Tm21_061, Tm21_087, Tm21_093, Tm21_100, Tm21_102, Tm21_104, Tm21_120, Tm21_139, Tm21_143, Tm21_157, Tm21_168, Tm21_182, Tm21_186, Tm21_239. The 31 clonal lines were randomly selected from a total of 240 plants. Phenotypic variation among these lines, based on plants grown from seed over two years, is presented in S2 Fig. This variation was visualized using hierarchical clustering of standardized compound contents.

To estimate broad-sense heritability of the medicinal compound contents, the stolons of 26 clonal lines (Tm21_013, Tm21_016, Tm21_020, Tm21_026, Tm21_028, Tm21_030, Tm21_035, Tm21_036, Tm21_045, Tm21_049, Tm21_051, Tm21_052, Tm21_055, Tm21_058, Tm21_087, Tm21_093, Tm21_100, Tm21_102, Tm21_104, Tm21_139, Tm21_143, Tm21_157, Tm21_168, Tm21_182, Tm21_186 and Tm21_239) were cut into 10 cm segments and planted in the experimental field. The plants were grown over a two-year period, from May 3, 2022, to November 20, 2023. Cultivation was carried out using three to ten biological replicates, and adventitious roots from each clonal line were weighed and used for the quantification of medicinal compounds. The root samples were also used to compare the medicinal compound

contents between the main roots from the plant grown from the seeds and the adventitious roots from the plant grown from the stolons.

To examine whether the evaluation period could be shortened, a comparison between one- and two-year cultivation was conducted using the following 14 clonal lines: Tm21_016, Tm21_028, Tm21_035, Tm21_036, Tm21_049, Tm21_055, Tm21_058, Tm21_087, Tm21_104, Tm21_139, Tm21_168, Tm21_182, Tm21_186, and Tm21_239. The clonal lines were cultivated from May 3, 2022, to November 20, 2023, for the two-year plants, and from April 28, 2023, to November 20, 2023, for the one-year plants. The harvested roots were used for the quantification of medicinal constituents.

To assess the environmental difference of cultivation year on the medicinal compound contents in *G. uralensis*, the following 13 clonal lines were developed in 2022 and 2023 for one year: Tm21_016, Tm21_028, Tm21_029, Tm21_035, Tm21_037, Tm21_060, Tm21_061, Tm21_087, Tm21_104, Tm21_120, Tm21_139, Tm21_186, and Tm21_239. The clonal lines were grown from May 3, 2022, to November 11, 2022, and from April 28, 2023, to November 20, 2023, each for one year. The plants were cultivated with three to ten biological replicates, and the medicinal compound contents in the roots were analyzed for each clonal line. Details of the clonal lines used in each experiment, along with an overview of the experimental design, are provided in S1 Fig.

## HPLC analysis of glycyrrhizin and flavonoids

The harvested roots of *G. uralensis* were dried in a convection oven at 50 °C for 3 days, and their dry weights were measured. The dried roots were pulverized using a vibrating mill (Cosmic Mechanical Technology, TI-200). Seven medicinal compounds, glycyrrhizin, liquiritin, liquiritin apioside, isoliquiritin, isoliquiritin apioside, liquiritigenin, and isoliquiritigenin (Fig 1), were quantified using High Performance Liquid Chromatography (HPLC) analysis as described below. Powdered samples of *G. uralensis* roots were precisely weighed to 0.125 g, extracted with 25 mL of 50% ethanol using a reciprocal shaker (Taitec, model SR-1) for 15 min, and then centrifuged at 1,660 *g* for 10 min. The supernatant was transferred to a 1.5 mL microtube, centrifuged at 13,800 × g for 5 min, and then used for HPLC analysis on an ACQUITY UPLC H-Class system (Waters, Milford, MA, USA) equipped with a photodiode array (PDA) detector. Chromatographic separation was achieved using an ACQUITY UPLC BEH C18 reversed-phase column (1.7 μm, 2.1 × 100 mm). The mobile phase consisted of water containing 0.7% acetic acid and 0.5% ammonium acetate (A) and acetonitrile (B). The gradient program for the mobile phase was as follows: 0–30 min, 5–15% B; 30–45 min, 15–45% B; 45–50 min, 95% B; stop at 50 min. The flow rate was maintained at 0.40 mL/min, with the column temperature set at 40 °C. Detection was performed at wavelengths

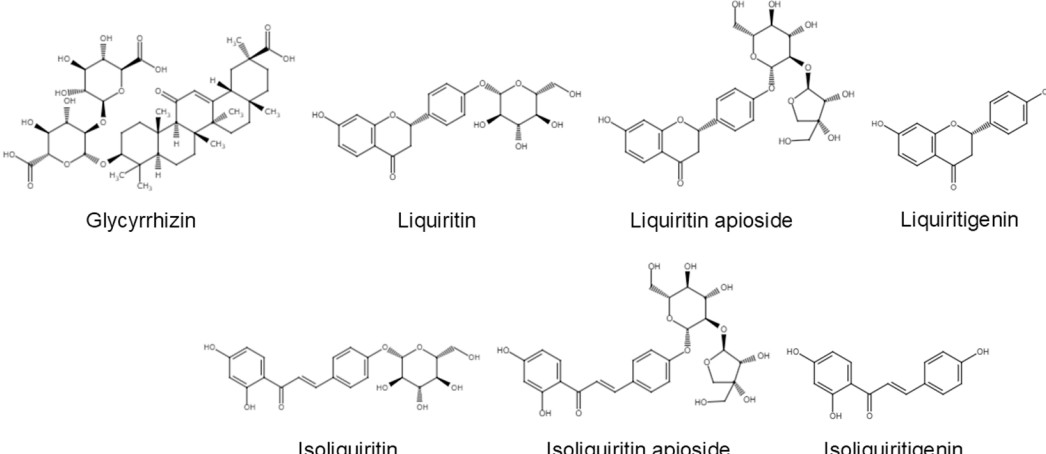

**Fig 1. Target chemical structure of the medicinal compounds in *G. uralensis*.**

of 254, 275, and 360 nm using the PDA detector. The identification of target compounds was conducted by comparing the retention times and UV spectra obtained from HPLC with those of the standard compounds. The compound contents in the samples were calculated based on the dry weights of the powdered materials.

## Statistical analysis

One-way ANOVA, two-way ANOVA, and correlation analysis using Pearson's correlation coefficient were performed to evaluate the heritability of glycyrrhizin, liquiritin, liquiritin apioside, isoliquiritin, isoliquiritin apioside, liquiritigenin, and isoliquiritigenin content in *G. uralensis*. The analyses were conducted using R (version 4.2.2). Broad-sense heritability ($h_B$) was estimated using variance components derived from ANOVA, according to the formula: $h_B = \sigma^2_G / (\sigma^2_G + \sigma^2_E)$, where $\sigma^2_G$ denotes the genotypic variance, and $\sigma^2_E$ denotes the environmental variance [25]. The effective number of replicates (r) for the estimation of $\sigma^2_G$ was calculated using the following equation: $r = ((\Sigma_{i=1}^n r_i - \Sigma_{i=1}^n r_i^2 / \Sigma_{i=1}^n r_i) / (a - 1))$, where a corresponds to the number of clonal lines [26].

## Results

### Estimation of broad-sense heritability of the medicinal compound contents in *G. uralensis*

To estimate the heritability of medicinal compound contents in *G. uralensis*, 26 clonal lines were cultivated over two years in the experimental field, and the medicinal compound contents of the individual plants were measured. The target constituents were glycyrrhizin, a triterpenoid, and flavonoid compounds including liquiritin, liquiritin apioside, liquiritigenin, isoliquiritin, isoliquiritin apioside, and isoliquiritigenin. Liquiritigenin is the aglycone of liquiritin and liquiritin apioside, while isoliquiritigenin is the aglycone of isoliquiritin and isoliquiritin apioside (Fig 1). The contents of seven medicinal compounds in 26 clonal lines are shown in Fig 2. As can be seen in the figure, the variation within the clonal lines in terms of the contents of glycyrrhizin, liquiritin, liquiritin apioside, isoliquiritin, and isoliquiritin apioside was smaller than the variation among lines (Fig 2A–C, E, F). In contrast, the within-line variation of the contents of liquiritigenin and isoliquiritigenin was greater compared to the between-line variation (Fig 2D, G). From these data, we conducted a one-way ANOVA and estimated the broad-sense heritability of the medicinal constituent contents. As a result, significant differences were observed among lines for all seven constituent contents (Table 1). In addition, broad-sense heritability of the medicinal constituent contents was estimated using the variance components from ANOVA. The heritability values of the following components were relatively high: glycyrrhizic acid (0.69), liquiritin (0.64), liquiritin apioside (0.86), isoliquiritin (0.64), and isoliquiritin apioside (0.89). In contrast, the heritability of liquiritigenin and isoliquiritigenin showed relatively low values of 0.26 and 0.19, respectively (Table 1).

Correlation analyses between the content of each compound and root weight were conducted, and no significant correlations with medicinal constituent contents were observed (Fig 3A–F), except in the case of isoliquiritigenin, and this correlation was relatively weak. The correlation between isoliquiritigenin and root weight was relatively weak (Fig 3G).

### Correlation of the medicinal compound contents between seed-derived main roots and stolon-derived adventitious roots

To assess the heritability of medicinal constituent contents in *G. uralensis*, we analyzed the compound contents of the main roots of 26 individuals cultivated from seeds for two years. Stolons were collected from each individual, propagated as stolon-derived seedlings, and cultivated for two years as 26 independent lines. The compound contents of the adventitious roots obtained from these lines were subsequently measured. Correlation analyses were conducted to evaluate the relationship between the compound contents of seed-derived main roots and stolon-derived adventitious roots across the 26 lines. As a result, significant correlations were observed in the contents of glycyrrhizin, liquiritin, liquiritin apioside, liquiritigenin, isoliquiritin, and isoliquiritin apioside between main and adventitious roots, with correlation coefficients of $r = 0.65$, 0.72, 0.86, 0.56, 0.73, and 0.85, respectively (Fig 4A–F). In particular, high correlations were observed with respect to

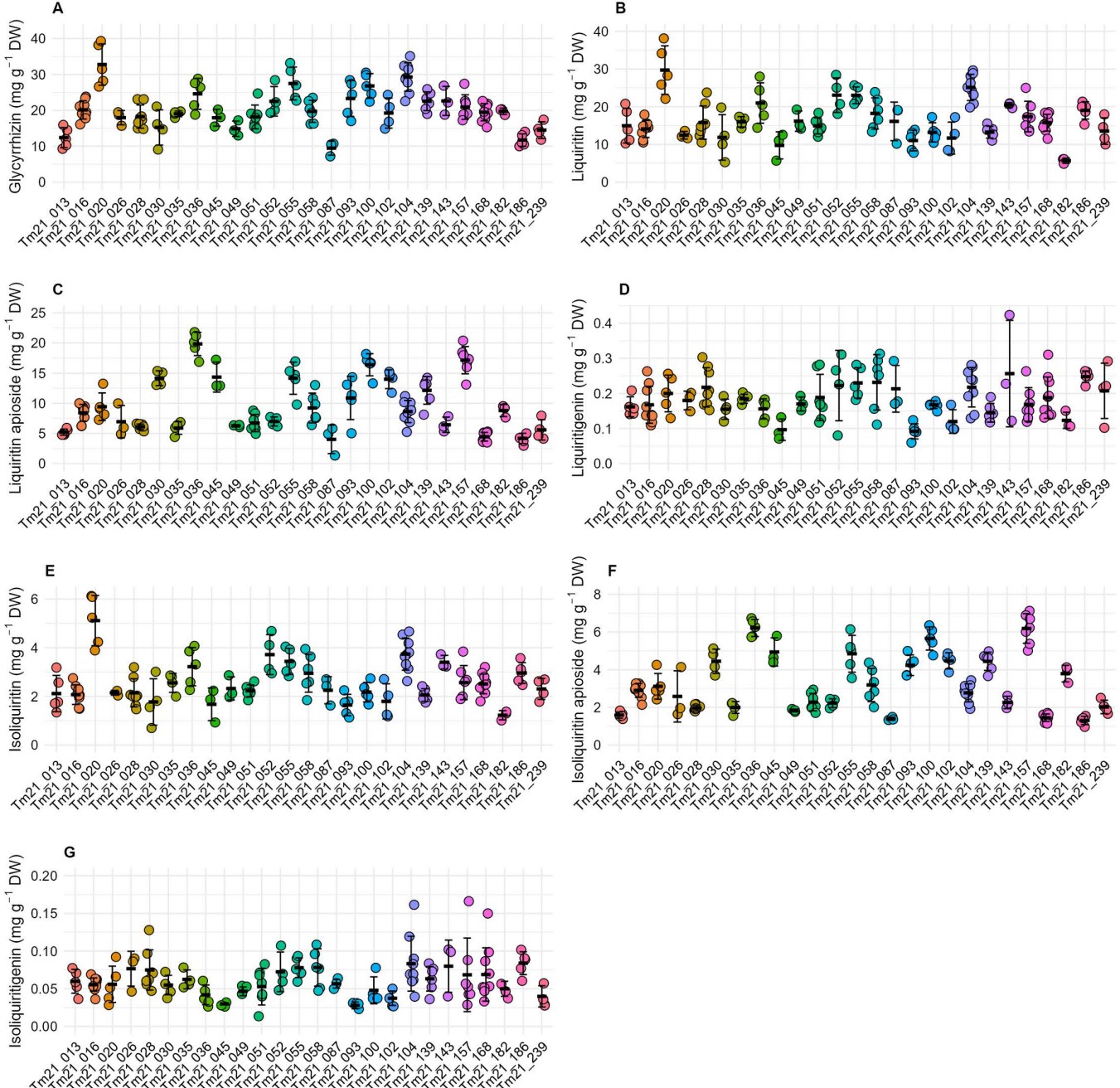

**Fig 2. Range of variation for chemical compound contents in *G. uralensis*. Each bars represent the mean ± SD.** Biological replicates within each clonal lines were as follows: Tm21_168 (n = 10), Tm21_016 and Tm21_104 (n = 9), Tm21_028 (n = 8), Tm21_051 and Tm21_157 (n = 7), Tm21_058 and Tm21_139 (n = 8), Tm21_013, Tm21_020, Tm21_036, Tm21_055, Tm21_093, Tm21_100 and Tm21_186 (n = 5), Tm21_030, Tm21_035, Tm21_052, Tm21_102 and Tm21_239 (n = 4), Tm21_026, Tm21_045, Tm21_049, Tm21_087, Tm21_143 and Tm21_182 (n = 3). Contents shown in panels A–G are: glycyrrhizin (A), liquiritin (B), liquiritin apioside (C), liquiritigenin (D), isoliquiritin (E), isoliquiritin apioside (F), and isoliquiritigenin (G).

**Table 1. Broad-sense heritability of the contents of medicinal compounds in *G. uralensis*.**

| Chemical compounds | Factors | Df | Mean Sq | P-value | Effective replication | Genotypic variance | Environmental variance | Broad sense heritability |
|---|---|---|---|---|---|---|---|---|
| Glycyrrhizin | Genotype (G) | 25 | 145.8 | <2.0E-16*** | 5.19 | 25.9 | 11.6 | 0.69 |
| | Residuals | 109 | 11.6 | | | | | |
| Liquiritin | Genotype (G) | 25 | 129.4 | <2.0E-16*** | 5.19 | 22.5 | 12.6 | 0.64 |
| | Residuals | 109 | 12.6 | | | | | |
| Liquiritin apioside | Genotype (G) | 25 | 101.5 | <2.0E-16*** | 5.19 | 19.0 | 3.2 | 0.86 |
| | Residuals | 109 | 3.2 | | | | | |
| Liquiritigenin | Genotype (G) | 25 | 8.4.E-03 | 9.3 E-05*** | 5.19 | 1.0.E-03 | 0.00 | 0.26 |
| | Residuals | 109 | 2.9.E-03 | | | | | |
| Isoliquiritin | Genotype (G) | 25 | 3.43 | <2.0E-16*** | 5.19 | 0.60 | 0.34 | 0.64 |
| | Residuals | 109 | 0.34 | | | | | |
| Isoliquiritin apioside | Genotype (G) | 25 | 12.2 | <2.0E-16*** | 5.19 | 2.30 | 0.29 | 0.89 |
| | Residuals | 109 | 0.29 | | | | | |
| Isoliquiritigenin | Genotype (G) | 25 | 1.3.E-03 | 3.0E-03* | 5.19 | 1.4.E-04 | 5.9.E-04 | 0.19 |
| | Residuals | 109 | 5.9.E-04 | | | | | |

Df; degree of freedom, Mean Sq; Mean square; ***Probability value for test of significance <0.001, *Probability value for test of significance <0.05.

liquiritin apioside and isoliquiritin apioside. In the case of liquiritigenin, although a significant correlation was observed between main roots and adventitious roots, the correlation coefficient was relatively low compared to what was found for the other compounds. Additionally, no significant correlation was observed between main roots and adventitious roots in the case of isoliquiritigenin content (Fig 4G).

## Effect of plant age on medicinal constituent contents

To determine whether the evaluation period for the medicinal compound contents in *G. uralensis* could be shortened, a comparison of medicinal constituent contents between one- and two-year plant ages was performed using 14 clonal lines, and the compound contents were analyzed using two-way ANOVA. The effects of genotype (G), plant age (A), and their interaction (G×A) were significant for all compound contents (Table 2). With each of the medicinal compounds, the mean squares of the G×A interaction were lower than those of the genotype and plant age. The mean square of plant age was higher than that of genotype for the contents of glycyrrhizin, liquiritin, liquiritin apioside, liquiritigenin, and isoliquiritin. In contrast, the mean square of plant age was lower than that of genotype in the case of isoliquiritin apioside and isoliquiritigenin contents. These results suggest that the relative contributions of genotype and plant age vary among the medicinal compounds, whereas their interaction effects were minimal.

A correlation analysis of the medicinal compound contents between one- and two-year plants was performed. Consequently, significant and strong correlations were observed in the contents of glycyrrhizin, liquiritin apioside, and isoliquiritin apioside between one- and two-year-old plants, with correlation coefficients (*r*) of 0.84, 0.87, and 0.84, respectively (Fig 5A, C, E). On the other hand, significant correlations were not observed in the contents of liquiritin (0.47), liquiritigenin (0.39), isoliquiritin (0.42), and isoliquiritigenin (0.28) (Fig 5B, D, E, G). These results suggest that the evaluation period for the medicinal compound contents in *G. uralensis* may be shortened.

## Effect of cultivation year on medicinal compound contents

To investigate the effect of cultivation year on the medicinal compound contents in *G. uralensis*, 13 clonal lines were cultivated in the same experimental field during 2022 and 2023. The compound contents were analyzed using two-way

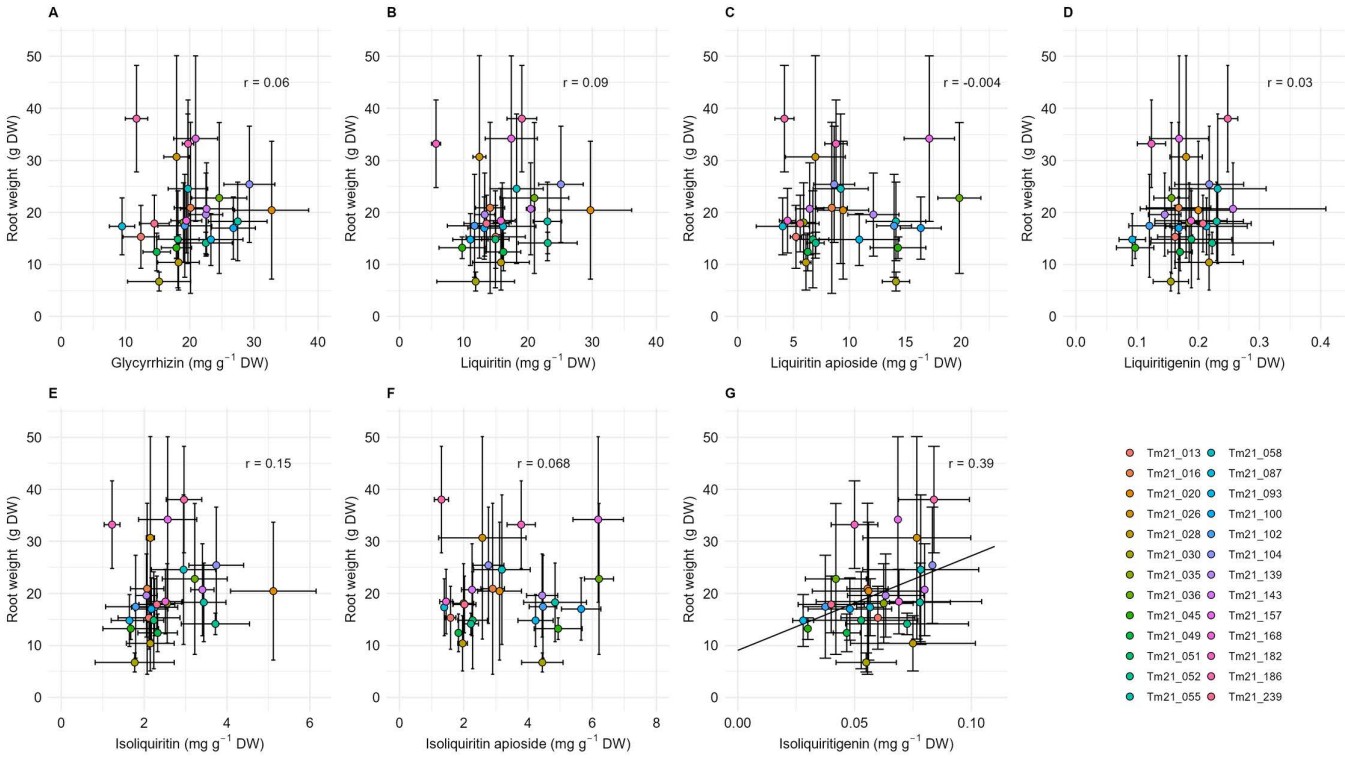

**Fig 3. Relationships between medicinal compound contents and rhizome weight in *G. uralensis*.** Each point represents the mean of 3–10 measurements and the bars indicate standard deviation (SD). Biological replicates within each clonal lines were as follows: Tm21_168 (n = 10), Tm21_016 and Tm21_104 (n = 9), Tm21_028 (n = 8), Tm21_051 and Tm21_157 (n = 7), Tm21_058 and Tm21_139 (n = 8), Tm21_013, Tm21_020, Tm21_036, Tm21_055, Tm21_093, Tm21_100 and Tm21_186 (n = 5), Tm21_030, Tm21_035, Tm21_052, Tm21_102 and Tm21_239 (n = 4), Tm21_026, Tm21_045, Tm21_049, Tm21_087, Tm21_143 and Tm21_182 (n = 3). The contents shown in A–G are: glycyrrhizin (A), liquiritin (B), liquiritin apioside (C), liquiritigenin, (D), isoliquiritin (E), isoliquiritin apioside (F), and isoliquiritigenin (G).

ANOVA. The effects of genotype (G) and cultivation year (Y) on glycyrrhizin content were significant; however, the G × Y interaction was not significant (Table 3). In the case of the other medicinal compounds, the effects of genotype (G), cultivation year (Y), and G × Y interaction on medicinal content were significant; however, the mean square for G × Y interactions was noticeably lower than that for genotype and cultivation year (Table 3). In addition, the mean square of the cultivation year was higher than that for the genotype, except in the case of liquiritin apioside (Table 3).

As indicated by *r* values (Fig 6), there was a moderate to high correlation between cultivation year and the contents of the following medicinal components: glycyrrhizin (0.87), liquiritin (0.68), liquiritin apioside (0.87), liquiritigenin (0.52), isoliquiritin (0.57), isoliquiritin apioside (0.88), and isoliquiritigenin (0.47). These correlations were statistically significant in five of the seven compounds: glycyrrhizin, liquiritin, liquiritin apioside, isoliquiritin, and isoliquiritin apioside (Fig 6A–C, E, F).

### Correlation analysis of contents of each medicinal compound

We conducted a correlation analysis of the contents of each medicinal compound using 26 clonal lines cultivated over two years in the experimental field. Moderate correlations (as indicated by *r* values) were observed between glycyrrhizin and four of the other compounds: liquiritin (0.52), liquiritin apioside (0.46), isoliquiritin (0.58), and isoliquiritin apioside (0.48). High correlations were found between liquiritin and isoliquiritin (0.97), as well as between liquiritin apioside and isoliquiritin

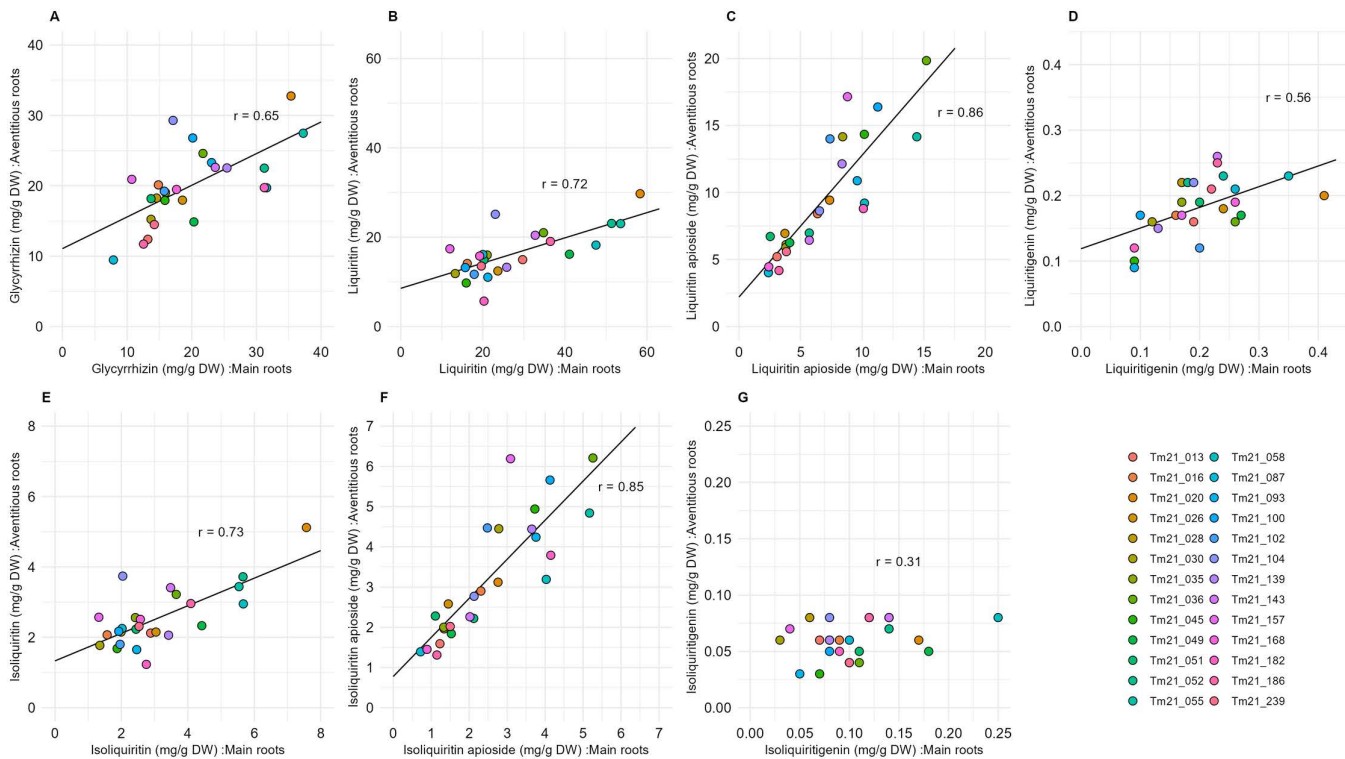

**Fig 4. Relationships between the contents of medicinal compounds in main roots and adventitious roots of *G. uralensis*.** The data represent the average content of each medicinal compound in adventitious roots, calculated based on 3 to 10 biological replicates. In contrast, the data for main roots are based on a single measurement (n = 1), with no replicates. Shown are the contents of: glycyrrhizin (A), liquiritin (B), liquiritin apioside contents (C), liquiritigenin (D), isoliquiritin (E), isoliquiritin apioside (F), isoliquiritigenin (G).

apioside (0.98). Moderate correlations were also observed between liquiritin and liquiritigenin, liquiritin and isoliquiritigenin, isoliquiritin and liquiritigenin, and isoliquiritin and isoliquiritigenin. The correlation coefficient between liquiritigenin and isoliquiritigenin was $r = 0.80$. In contrast, moderate negative correlations were observed between liquiritin apioside and liquiritigenin, as well as between isoliquiritin apioside and liquiritigenin, with correlation coefficients ($r$) of −0.47 and −0.52, respectively.

## Discussion

We determined that there were significant differences between the medicinal compound contents of different clonal lines of *Glycyrrhiza uralensis* grown in an experimental field. High heritability was observed for the contents of glycyrrhizin, liquiritin, liquiritin apioside, isoliquiritin, and isoliquiritin apioside (Table 1). These results indicate that selective breeding is effective in controlling the contents of medicinal compounds in *G. uralensis*, and high selective effects can be expected, particularly for glycyrrhizin, liquiritin, liquiritin apioside, isoliquiritin, and isoliquiritin apioside. Additionally, there were no significant correlations, or only weak correlations, between the contents of medicinal compounds and root weight (Fig 3). This indicates the potential to develop lines with high compound contents independently of root yield. In summary, selective breeding could facilitate the development of *G. uralensis* cultivars with both high medicinal compound contents and high yields.

Correlation analysis of the medicinal compound contents between seed-derived main roots and stolon-derived adventitious roots revealed high correlations in the contents of glycyrrhizin, liquiritin, liquiritin apioside, isoliquiritin, and isoliquiritin

**Table 2. Two-way ANOVA of the medicinal compound contents in first- and second-year *G. uralensis* plants.**

| Chemical compounds | Factors | Df | Mean Square | P-value | |
|---|---|---|---|---|---|
| **Glycyrrhizin** | Genotype (G) | 13 | 223.6 | <2.0E-16 | *** |
| | Plant age (A) | 1 | 1477.3 | <2.0E-16 | *** |
| | G×A | 13 | 37.1 | 3.1E-08 | *** |
| | Residuals | 160 | 6.7 | | |
| **Liquiritin** | Genotype (G) | 13 | 222.7 | <2.0E-16 | *** |
| | Plant age (A) | 1 | 2022.9 | <2.0E-16 | *** |
| | G×A | 13 | 64.4 | 1.1E-11 | *** |
| | Residuals | 160 | 8.4 | | |
| **Liquiritin apioside** | Genotype (G) | 13 | 111.7 | <2.0E-16 | *** |
| | Plant age (A) | 1 | 243.0 | <2.0E-16 | *** |
| | G×A | 13 | 23.2 | 1.5E-14 | *** |
| | Residuals | 160 | 2.4 | | |
| **Liquiritigenin** | Genotype (G) | 13 | 0.12 | <2.0E-16 | *** |
| | Plant age (A) | 1 | 0.60 | 1.8E-15 | *** |
| | G×A | 13 | 0.08 | 4.1E-16 | *** |
| | Residuals | 160 | 0.01 | | |
| **Isoliquiritin** | Genotype (G) | 13 | 4.9 | <2.0E-16 | *** |
| | Plant age (A) | 1 | 29.2 | <2.0E-16 | *** |
| | G×A | 13 | 1.5 | 2.8E-09 | *** |
| | Residuals | 160 | 0.2 | | |
| **Isoliquiritin apioside** | Genotype (G) | 13 | 15.2 | <2.0E-16 | *** |
| | Plant age (A) | 1 | 3.3 | 0.003 | ** |
| | G×A | 13 | 2.4 | 3.7E-10 | *** |
| | Residuals | 160 | 0.4 | | |
| **Isoliquiritigenin** | Genotype (G) | 13 | 0.006 | 8.0E-16 | *** |
| | Plant age (A) | 1 | 0.004 | 0.013 | * |
| | G×A | 13 | 0.004 | 3.1E-11 | *** |
| | Residuals | 160 | 0.001 | | |

Df; degree of freedom, Mean Sq; Mean square. *, **; ***; Probability value for test of significance <0.05, <0.01 and <0.001, respectively.

apioside (Fig 4A–C, E, F). These results show a similar trend to those observed in the broad-sense heritability analysis and suggest that the contents of glycyrrhizin, liquiritin, liquiritin apioside, isoliquiritin, and isoliquiritin apioside exhibit strong genetic inheritance. In a number of plant species, such as *Artemisia annua* [27], *Litchi chinensis*, and *Triticum aestivum* [28], the broad-sense heritability of terpenoid and flavonoid content has been shown to be high [29]. This suggests that the content of specific metabolites in plants exhibits strong heritability across plant species. On the other hand, in the present study, there were also specific metabolites with low heritability, such as liquiritigenin and isoliquiritigenin (Table 1). These metabolites are intermediates in the biosynthetic pathway of liquiritin and isoliquiritin and thus are considered less likely to accumulate [30]. It is reasonable to conclude that the low heritability of these two metabolites is due to this characteristic.

To evaluate the feasibility of early selection for strains with high levels of medicinal compounds, we compared the medicinal compound content in one- and two-year-old plants. The two-way ANOVA of genotype and plant age revealed that, in the case of glycyrrhizin and liquiritin, the mean square for plant age was greater than that for genotype (Table 2). However, the interaction between genotype and plant age was small, and correlation analysis showed a high to moderate

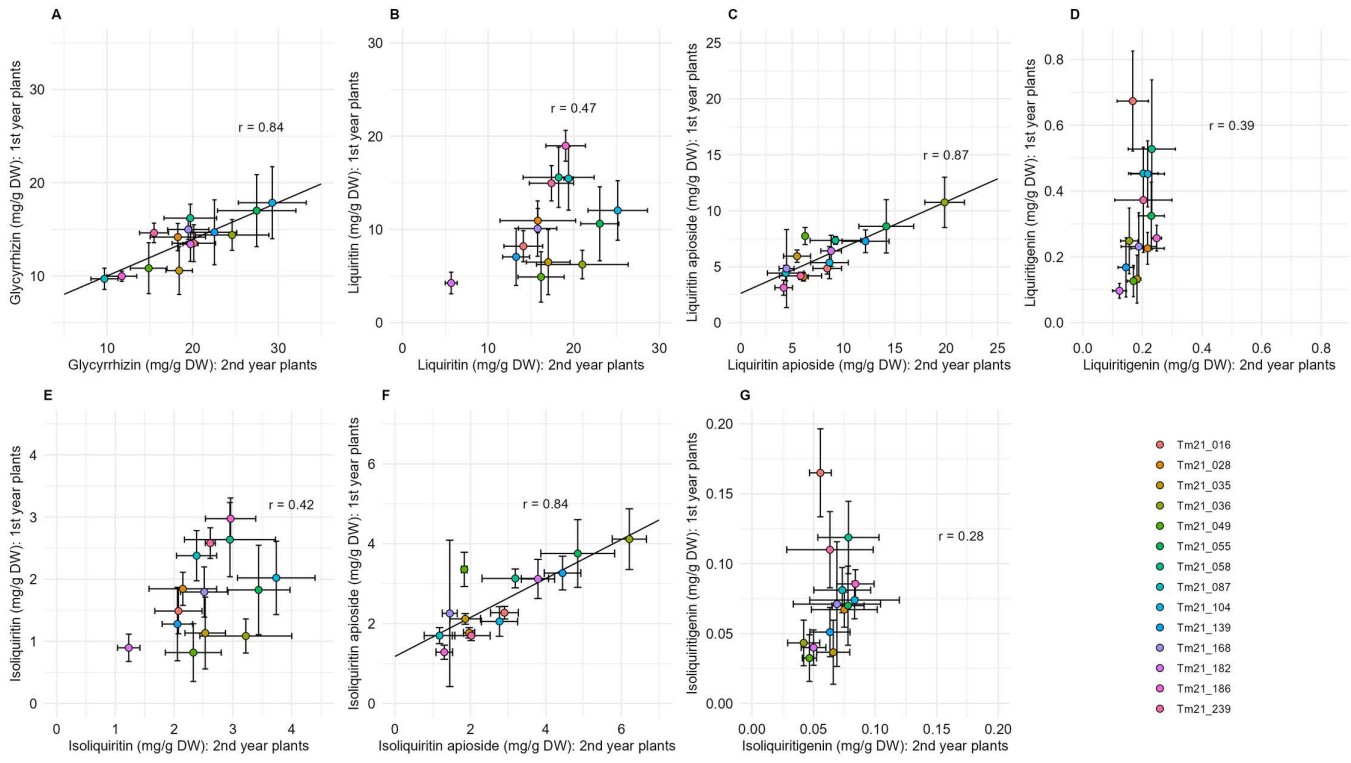

**Fig 5. Relationships between the contents of medicinal compounds in first- and second-year *G. uralensis* plants.** Each point represents the mean of 3–10 biological replicates, and the bars indicate the standard deviation (SD). Biological replicates for each clonal line were as follows: for first-year plants, Tm21_016 (n = 9), Tm21_028 (n = 8), Tm21_035 (n = 5), Tm21_036 (n = 5), Tm21_049 (n = 3), Tm21_055 (n = 5), Tm21_058 (n = 6), Tm21_087 (n = 3), Tm21_104 (n = 9), Tm21_139 (n = 6), Tm21_168 (n = 10), Tm21_182 (n = 3), Tm21_186 (n = 5) and Tm21_239 (n = 3); for second-year plants, Tm21_016 (n = 6), Tm21_028 (n = 7), Tm21_035 (n = 9), Tm21_036 (n = 6), Tm21_049 (n = 8), Tm21_055 (n = 9), Tm21_058 (n = 8), Tm21_087 (n = 9), Tm21_104 (n = 5), Tm21_139 (n = 9), Tm21_168 (n = 9), Tm21_182 (n = 6), Tm21_186 (n = 9) and Tm21_239 (n = 8).

correlation in compound content between one- and two-year-old plants (Table 2, Fig 5). These results indicate that while the absolute content of medicinal compounds varies significantly with plant age, the relative differences between strains are small. In other words, strains that exhibit high compound content in one-year-old plants are likely to show similarly high content compared to other strains in two-year-old plants. Therefore, it is considered feasible to evaluate and select for medicinal compound content in *G. uralensis* at an early stage, such as in one-year-old plants. In other plant species, such as *Sinopodophyllum hexandrum* and *Betula platyphylla*, it has been reported that flavonoid and triterpene biosynthesis-related enzyme genes are upregulated with plant age, along with an increase in their contents [31,32]. Similarly, it is possible that the absolute content of medicinal compounds in *G. uralensis* also increases with the upregulation of gene expression as the plant ages. However, further research is needed in this regard.

We also evaluated the environmental effects of cultivation year on the content of medicinal compounds in *G. uralensis*. A two-way ANOVA of genotype and cultivation year revealed that, for most medicinal compounds except liquiritin apioside, the mean square for cultivation year was larger than that for genotype (Table 3). However, the interaction between genotype and cultivation year was minimal, and correlation analysis showed a high to moderate correlation in compound content across the two cultivation years, except in the case of liquiritigenin and isoliquiritigenin (Table 3, Fig 6). These results indicate that, although *G. uralensis* was cultivated in the same experimental field, environmental variations across different cultivation years caused significant changes in the absolute content of medicinal compounds, while the relative

**Table 3. Two-way ANOVA of the medicinal compound contents in first- and second-year *G. uralensis* plants.**

| Chemical compounds | Factors | Df | Mean Square | P-value | |
|---|---|---|---|---|---|
| Glycyrrhizin | Genotype (G) | 12 | 283.7 | <2.0E-16 | *** |
| | Year (Y) | 1 | 507.9 | 1.3E-12 | *** |
| | G × Y | 12 | 13.7 | 0.0812 | |
| | Residuals | 128 | 8.2 | | |
| Liquiritin | Genotype (G) | 12 | 184.5 | <2.0E-16 | *** |
| | Year (Y) | 1 | 1832.4 | <2.0E-16 | *** |
| | G × Y | 12 | 31.2 | 9.6E-05 | *** |
| | Residuals | 128 | 8.5 | | |
| Liquiritin apioside | Genotype (G) | 12 | 63.0 | <2.0E-16 | *** |
| | Year (Y) | 1 | 37.9 | 4.0E-07 | *** |
| | G × Y | 12 | 3.4 | 0.00453 | ** |
| | Residuals | 128 | 1.3 | | |
| Liquiritigenin | Genotype (G) | 12 | 2.0 | <2.0E-16 | *** |
| | Year (Y) | 1 | 218.9 | <2.0E-16 | *** |
| | G × Y | 12 | 0.8 | 1.7E-11 | *** |
| | Residuals | 128 | 0.1 | | |
| Isoliquiritin | Genotype (G) | 12 | 4.4 | <2.0E-16 | *** |
| | Year (Y) | 1 | 12.7 | 1.6E-10 | *** |
| | G × Y | 12 | 7.0 | <2.0E-16 | *** |
| | Residuals | 128 | 0.3 | | |
| Isoliquiritin apioside | Genotype (G) | 12 | 2.8 | <2.0E-16 | *** |
| | Year (Y) | 1 | 152.7 | <2.0E-16 | *** |
| | G × Y | 12 | 2.5 | <2.0E-16 | *** |
| | Residuals | 128 | 0.16 | | |
| Isoliquiritigenin | Genotype (G) | 12 | 0.020 | <2.0E-16 | *** |
| | Year (Y) | 1 | 0.69 | <2.0E-16 | *** |
| | G × Y | 12 | 0.007 | 1.7E-11 | *** |
| | Residuals | 128 | 0.001 | | |

Df; degree of freedom, Mean Sq; Mean square. *, **; ***; Probability value for test of significance <0.05, <0.01 and <0.001, respectively.

ranking of medicinal compound content among strains remained largely stable. In a study by Ramilowski *et al*., gene expression related to glycyrrhizin biosynthesis was compared in high- and low-glycyrrhizin-producing strains; the authors reported that the expression levels of biosynthesis enzyme genes, such as *CYP88D6* and *CYP72A154*, were relatively higher in the high-producing strains [33]. In the present study, although the absolute content of compounds changedto environmental influences, the relative relationship of compound content between strains remained unchanged, which may be attributed to the expression levels of related genes.

In general, secondary metabolites, such as triterpenes and flavonoids, are produced in response to biotic and abiotic stress via phytohormone signaling pathways [34]. It has been reported that glycyrrhizin and liquiritin accumulate in *G. uralensis* under drought and salt stress through abscisic acid (ABA) and jasmonic acid (JA) signaling pathways [21,35]. Additionally, it has been revealed that low temperatures in the cultivation environment increase the glycyrrhizin content [16]. In the present study, the average temperature of cultivation periods in 2022 was lower than that in 2023 (S1 Table), and the contents of medicinal compounds in 2022 were higher than those in 2023 (Fig 6). These environmental variations

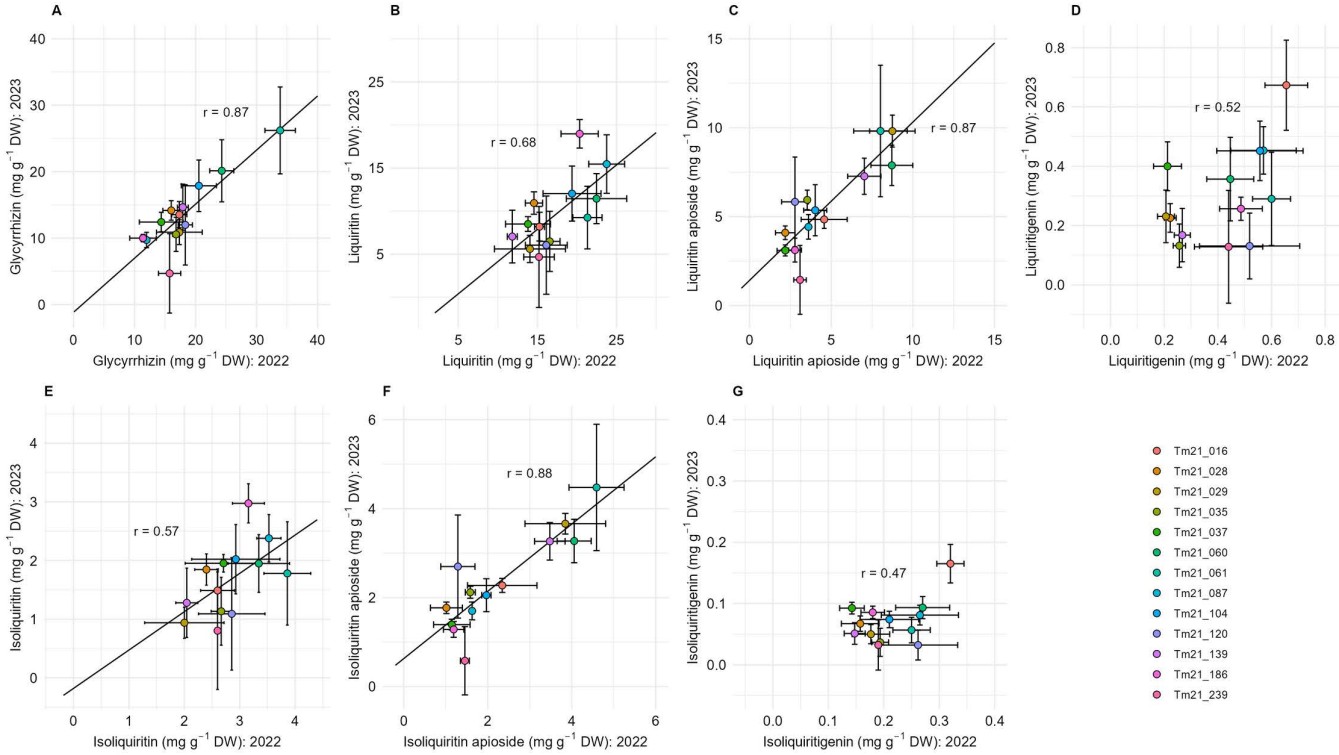

**Fig 6. Relationships between the contents of medicinal compounds from 13 clonal lines of *G. uralensis* grown in 2022 and 2023.** Each point represents the mean of 3–10 biological replicates, and the bars indicate the standard deviation (SD). The number of biological replicates for each clonal line was as follows: for plants grown in 2022, Tm21_016 (n=4), Tm21_028 (n=4), Tm21_029 (n=6), Tm21_035 (n=3), Tm21_037 (n=4), Tm21_060 (n=8), Tm21_061 (n=4), Tm21_087 (n=4), Tm21_104 (n=3), Tm21_120 (n=6), Tm21_139 (n=4), Tm21_186 (n=4) and Tm21_239 (n=3); for plants grown in 2023, Tm21_016 (n=6), Tm21_028 (n=7), Tm21_029 (n=10), Tm21_035 (n=9), Tm21_037 (n=4), Tm21_060 (n=9), Tm21_061 (n=3), Tm21_087 (n=9), Tm21_104 (n=5), Tm21_120 (n=9), Tm21_139 (n=9), Tm21_186 (n=9) and Tm21_239 (n=8).

may have influenced the compound content, but further detailed investigation is needed to determine which specific environmental factors affect compound content.

Lastly, we conducted a correlation analysis among the contents of each medicinal compound and observed significant correlations between certain compound contents. Moderate correlations were observed between glycyrrhizin and certain flavonoids—liquiritin, liquiritin apioside, isoliquiritin, and isoliquiritin apioside—despite glycyrrhizin and flavonoids being synthesized through different biosynthetic pathways (Fig 7). The correlation between glycyrrhizin and liquiritin has been reported in previous studies [15]. Recent genomic analyses have revealed that genes related to the biosynthetic pathways of glycyrrhizin and flavonoids are clustered. However, while glycyrrhizin biosynthesis-related genes are clustered on chromosome 1, flavonoid biosynthesis-related genes are located on different chromosomes [18]. Thus, the detailed reason for the positive correlation between the contents of glycyrrhizin and flavonoids is not yet clear at this point, and further investigations using molecular analysis are needed. In any case, it is important to note that these traits are likely to be linked, which should be considered when developing strategies for improved breeding of *G. uralensis* and possibly other medicinal plants.

The correlation coefficients between liquiritin and isoliquiritin, and between liquiritin apioside and isoliquiritin apioside, were 0.97 and 0.98, respectively (Fig 7). The glycosyltransferase involved in the biosynthetic steps converting liquiritigenin to liquiritin and isoliquiritigenin to isoliquiritin is encoded by the same gene (*GuGT53*). Similarly, the apiosyltransferase

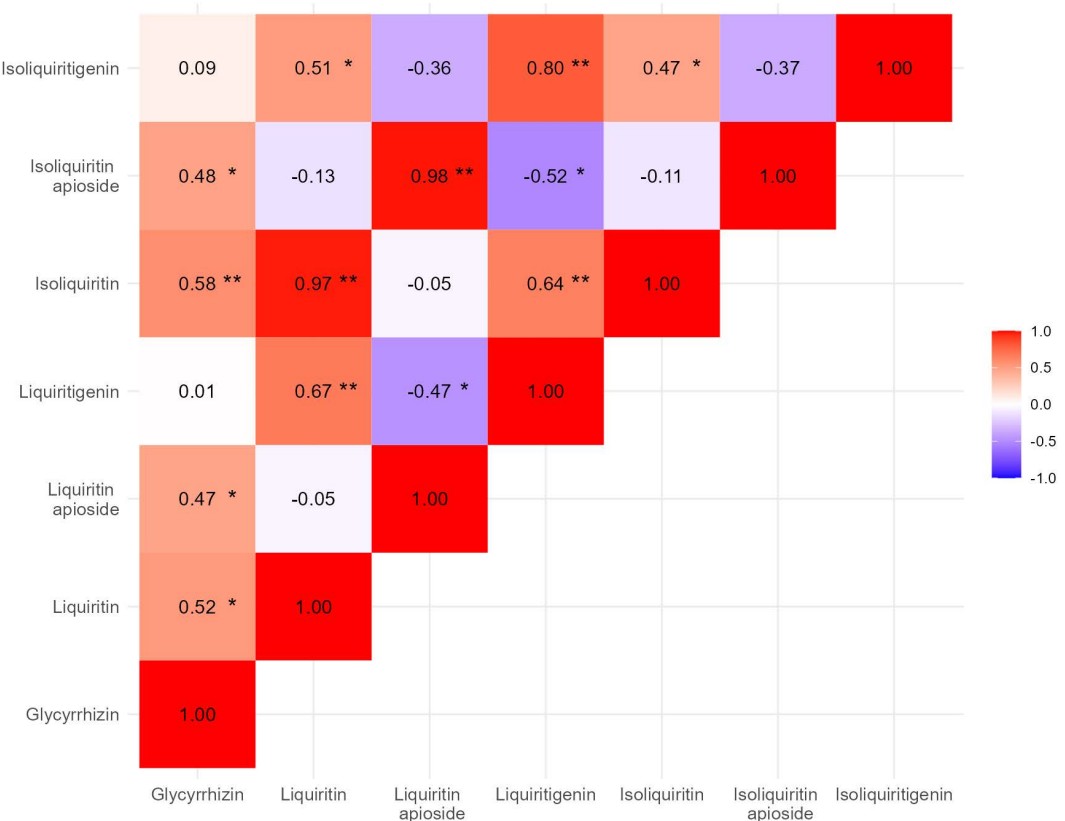

**Fig 7. Heat map for correlation analysis among each medicinal compound contents in *G. uralensis*.** The data were obtained from 24 clonal lines cultivated over two years, from 2022 to 2023. The correlation coefficient values range from −1 to 1, where positive values indicate a positive correlation, and negative values indicate a negative correlation. Blue indicates negative correlations, red indicates positive correlations, and the intensity of the color represents the strength of the correlation. *, **; Probability value for test of significance <0.05 and < 0.01, respectively.

responsible for the conversion of liquiritin to liquiritin apioside and isoliquiritin to isoliquiritin apioside is encoded by the same gene (*GuApiGT*) [36]. The high correlation we observed in the contents of these medicinal compounds is likely attributed to their biosynthesis being governed by the same gene. In addition, significant correlations were observed among liquiritin, isoliquiritin, liquiritigenin, and isoliquiritigenin (Fig 7), which are considered to exhibit high content correlations due to their closely related biosynthetic pathways [36].

In conclusion, we evaluated the genetic inheritance of the contents of seven medicinal compounds in *G. uralensis* using several approaches and observed high heritability for the contents of glycyrrhizin, liquiritin, liquiritin apioside, isoliquiritin, and isoliquiritin apioside. This suggests that selective breeding could be an effective strategy for developing *G. uralensis* populations with high medicinal compound contents. In addition, no correlation was observed between the contents of medicinal compounds and root yield. Consequently, *G. uralensis* cultivars with both high root yields and high medicinal compound contents could be developed. However, correlations among the medicinal compound contents were observed; therefore, it is important to consider the linkage between compounds when conducting selective breeding. The number of replicates per clonal line varied from 3 to 10 due to the instability in cultivating *G. uralensis*. In addition, because *G. uralensis* is self-incompatible and cannot be genetically fixed [11], the number of replicates for seed-derived main roots was limited to one. Although this variation in sample size may have reduced the statistical power of the ANOVA, the consistent trends in medicinal compound contents observed across multiple analyses support the validity of the main conclusions

of this study. Glycyrrhizin and flavonoid compounds possess strong pharmacological activities. In addition, glycyrrhizin is widely used as a natural sweetener and in cosmetic formulations [2]. Therefore, the simultaneous improvement of medicinal compound content and root yield in *G. uralensis* cultivars holds significant potential for enhancing their commercial and industrial value. We believe these results contribute significantly to decision-making processes in breeding programs and the development of further foundational studies of *G. uralensis*.

## Supporting information

**S1 Fig. Overview of the experimental design using clonal lines of G. uralensis to evaluate the heritability of medicinal compound contents.** (1) A total of 240 seed-derived plants were cultivated over two years (2020–2021), and 31 clonal lines were established via stolon propagation. (2) Twenty-six clonal lines were cultivated for two years (2022–2023) using stolon-derived seedlings to estimate broad-sense heritability and analyze correlations among medicinal compound contents. (3) Thirteen clonal lines were cultivated for one year in both 2022 and 2023 to assess year-to-year variation. (4) Fourteen clonal lines were cultivated for one year in 2023 to compare one- and two-year-old plants. Stolon-derived seedlings were used in all experiments except in (1). Roots were sampled and analyzed for medicinal compound contents using HPLC.
(PDF)

**S2 Fig. Hierarchical clustering of 31 clonal lines based on standardized contents of seven medicinal compounds.** To facilitate comparison across compounds with different scales, all compound content values were standardized using Z-score normalization. Specifically, each value was transformed by subtracting the mean and dividing by the standard deviation for the corresponding compound, resulting in a distribution with a mean of 0 and a standard deviation of 1. Hierarchical clustering was subsequently performed on the standardized data to visualize phenotypic variation among the 31 clonal lines. Clustering was conducted based on Euclidean distance and Ward's linkage method.
(PDF)

**S1 Table. Climatic conditions around the research field where *G. uralensis* was cultivated in 2022 and 2023.**
(PDF)

## Acknowledgments

We are grateful to Yoichi Aoki for technical support in the quantification of medicinal compound contents. We also thank Terue Kurosawa for cultivation assistance. We would like to thank Editage (www.editage.jp) for English language editing.

## Author contributions

**Conceptualization:** Takahiro Tsusaka.

**Data curation:** Takahiro Tsusaka.

**Formal analysis:** Takahiro Tsusaka.

**Investigation:** Takahiro Tsusaka, Miki Sakurai.

**Methodology:** Takahiro Tsusaka.

**Project administration:** Takahiro Tsusaka.

**Resources:** Takahiro Tsusaka, Miki Sakurai.

**Supervision:** Takahiro Tsusaka, Miki Sakurai.

**Validation:** Takahiro Tsusaka.

**Visualization:** Takahiro Tsusaka.

**Writing – original draft:** Takahiro Tsusaka.

**Writing – review & editing:** Takahiro Tsusaka, Miki Sakurai.

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
