## [Decision Letter · Decision Letter 0]

27 May 2025

PONE-D-25-10815Evaluation of genetic inheritance of the medicinal compound contents in Glycyrrhiza uralensisPLOS ONE

Dear Dr. Tsusaka,

Thank you for submitting your manuscript to PLOS ONE. After careful consideration, we feel that it has merit but does not fully meet PLOS ONE’s publication criteria as it currently stands. Therefore, we invite you to submit a revised version of the manuscript that addresses the points raised during the review process.

We look forward to receiving your revised manuscript.

Kind regards,

Mehdi Rahimi, Ph.D.

Academic Editor

PLOS ONE

Journal Requirements:

2. We note that your Data Availability Statement is currently as follows: All relevant data are within the manuscript and in Supporting Information files.

Reviewers' comments:

Reviewer's Responses to Questions

**Comments to the Author**

1. Is the manuscript technically sound, and do the data support the conclusions?

Reviewer #1: Yes

Reviewer #2: Partly

2. Has the statistical analysis been performed appropriately and rigorously? 

Reviewer #1: Yes

Reviewer #2: Yes

3. Have the authors made all data underlying the findings in their manuscript fully available?

Reviewer #1: Yes

Reviewer #2: Yes

4. Is the manuscript presented in an intelligible fashion and written in standard English?

Reviewer #1: Yes

Reviewer #2: Yes

5. Review Comments to the Author

Reviewer #1: The authors have carried quite detailed amount of work and carried out large experimentation on the genetic inheritance of the contents of seven medicinal compounds in G. uralensis and observed high heritability for the contents of glycyrrhizin, liquiritin, liquiritin apioside, isoliquiritin, and isoliquiritin apioside. This suggests that selective breeding could be an effective strategyfor developing G. uralensis populations with high medicinal compound contents. G. uralensis cultivars with

both high root yields and high medicinal compound contents could be developed and utilzed for commercial application. It is advised that kindly highlight the importance of these metabolites in the available herbal drugs or any other formulations used in their market so that in the future course of time,those newly breeding lines could be grown.

Reviewer #2: This manuscript by Takahiro Tsusaka and Miki Sakurai described that they investigated the genetic inheritance (heritability) of medicinal compound contents in G. uralensis by analyzing clonal lines propagated from stolons. And they conclude that there is strong potential for selective breeding to enhance the medicinal quality of G. uralensis by targeting high-yielding genotypes with superior compound profiles and that this potential may also apply to the sustainable cultivation of other high-quality medicinal plant resources. The work is interesting. But there are still some parts need to be enhanced.

1. In the parts of plant materials and cultivation, the relationship among the materials used in the different experiments is not clear enough, especially when comparing the status of annual and biennial G. uralensis (Is the annual liquorice material a subset of the biennial? ), and could be supplemented with a designed figure or table to illustrate this issue?

2. In the parts of plant materials and cultivation, Please provide the latitude and longitude information of the experimental site.

3. Sample size varies considerably among the clonal lines, from 3 to 9, and it is recommended that the data be supplemented to a sample size of 5 or more.

4. Genetic analysis using clonal lines requires sufficient genetic distance among different lines. However, the 26 clonal lines in this study came from the same batch of seeds, and how about the genetic distance between them? So experimental evidence of how genetically distant they are from each other is requested.

5. The data for the primary root in Figure 4, ‘Correlation of primary root and adventitious root components’, were measured only once (n=1). It is recommended to add repeated measurements of primary roots or to justify the single measurement.

6. Figure 1 (compound structure) is not cited in the main text. It is recommended that it be mentioned in the methods or results section or that Figure 1 be removed from the text.

6. PLOS authors have the option to publish the peer review history of their article (what does this mean? ). If published, this will include your full peer review and any attached files.

**Do you want your identity to be public for this peer review?** For information about this choice, including consent withdrawal, please see our Privacy Policy .

Reviewer #1: No

Reviewer #2: No

---

## [Author Response · Author response to Decision Letter 1]

17 Jun 2025

Dear Reviewers

Thank you for the thoughtful and constructive feedback you provided regarding our manuscript. We agree with you and have incorporated these suggestions throughout our paper. Detailed responses on your suggestions are as follows:

Reviewer #1’s comments

Reviewer #1: The authors have carried quite detailed amount of work and carried out large experimentation on the genetic inheritance of the contents of seven medicinal compounds in G. uralensis and observed high heritability for the contents of glycyrrhizin, liquiritin, liquiritin apioside, isoliquiritin, and isoliquiritin apioside. This suggests that selective breeding could be an effective strategy for developing G. uralensis populations with high medicinal compound contents. G. uralensis cultivars with both high root yields and high medicinal compound contents could be developed and utilzed for commercial application. It is advised that kindly highlight the importance of these metabolites in the available herbal drugs or any other formulations used in their market so that in the future course of time, those newly breeding lines could be grown.

Response to Reviewer #1

Thank you very much for your constructive feedback on our manuscript. We fully agree with your suggestion. In response, we have added a description of the industrial importance of the medicinal compounds in lines 422–425 of the Discussion section.

Reviewer #2’s comments

Reviewer #2: This manuscript by Takahiro Tsusaka and Miki Sakurai described that they investigated the genetic inheritance (heritability) of medicinal compound contents in G. uralensis by analyzing clonal lines propagated from stolons. And they conclude that there is strong potential for selective breeding to enhance the medicinal quality of G. uralensis by targeting high-yielding genotypes with superior compound profiles and that this potential may also apply to the sustainable cultivation of other high-quality medicinal plant resources. The work is interesting. But there are still some parts need to be enhanced.

1. In the parts of plant materials and cultivation, the relationship among the materials used in the different experiments is not clear enough, especially when comparing the status of annual and biennial G. uralensis (Is the annual liquorice material a subset of the biennial? ), and could be supplemented with a designed figure or table to illustrate this issue?

2. In the parts of plant materials and cultivation, Please provide the latitude and longitude information of the experimental site.

3. Sample size varies considerably among the clonal lines, from 3 to 9, and it is recommended that the data be supplemented to a sample size of 5 or more.

4. Genetic analysis using clonal lines requires sufficient genetic distance among different lines. However, the 26 clonal lines in this study came from the same batch of seeds, and how about the genetic distance between them? So experimental evidence of how genetically distant they are from each other is requested.

5. The data for the primary root in Figure 4, ‘Correlation of primary root and adventitious root components’, were measured only once (n=1). It is recommended to add repeated measurements of primary roots or to justify the single measurement.

6. Figure 1 (compound structure) is not cited in the main text. It is recommended that it be mentioned in the methods or results section or that Figure 1 be removed from the text.

Response to Reviewer #2

Thank you very much for your valuable and insightful comments. We have carefully considered each point and revised the manuscript accordingly. Our detailed point-by-point responses are provided below.

Comment 1: In the parts of plant materials and cultivation, the relationship among the materials used in the different experiments is not clear enough, especially when comparing the status of annual and biennial G. uralensis (Is the annual liquorice material a subset of the biennial? ), and could be supplemented with a designed figure or table to illustrate this issue?

Response: We have added a schematic figure illustrating the experimental design and the plant materials used in each experiment, which is included in Appendix S1.

Comment 2: In the parts of plant materials and cultivation, Please provide the latitude and longitude information of the experimental site.

Response: We have added the latitude and longitude information of the experimental site in lines 107–108 of the Materials and Methods section.

Comment 3: Sample size varies considerably among the clonal lines, from 3 to 9, and it is recommended that the data be supplemented to a sample size of 5 or more.

Response: Due to the difficulty of cultivating G. uralensis, we were unable to maintain uniform replication numbers across all clonal lines. Since the plant requires several years to grow, it is not feasible to increase the number of samples at this stage of the study.

However, we recognize that a sample size of five or more is desirable to ensure adequate statistical power for variance analysis. Therefore, we re-estimated the broad-sense heritability values using only the 15 clonal lines with five or more replicates and compared the results to those obtained using all 26 lines with three or more replicates. The heritability estimates based on the 15 lines were as follows: glycyrrhizin 0.69, liquiritin 0.64, liquiritin apioside 0.86, liquiritigenin 0.29, isoliquiritin 0.66, isoliquiritin apioside 0.91, and isoliquiritigenin 0.15. These values were nearly identical to those obtained from the dataset of 26 clonal lines, indicating that the use of lines with three or more replicates still yields reliable results.

Accordingly, we have clearly stated this limitation in lines 418–422 of the Discussion section and explained that, based on this consideration, we chose to proceed with the analysis using clonal lines with three or more replicates.

Comment 4: Genetic analysis using clonal lines requires sufficient genetic distance among different lines. However, the 26 clonal lines in this study came from the same batch of seeds, and how about the genetic distance between them? So experimental evidence of how genetically distant they are from each other is requested.

Response: Since genotyping would require cultivating additional plants, which takes several years, it is difficult to include such data in the current study. To provide some support for the genetic diversity among the clonal lines, we have added a statement in lines 115–116 of the Materials and Methods section noting that the clonal lines were randomly selected from a population of 240 individuals. In addition, we have included Appendix S2, which presents phenotypic variation in compound contents and a clustering analysis based on these traits among the 31 clonal lines used in the experiments. We plan to conduct a genotypic analysis in a subsequent study and report the results in a future publication.

Comment 5: The data for the primary root in Figure 4, ‘Correlation of primary root and adventitious root components’, were measured only once (n=1). It is recommended to add repeated measurements of primary roots or to justify the single measurement.

Response: Due to the biological characteristics of G. uralensis, which is highly self-incompatible and genetically heterogeneous, it is not possible to obtain genetically identical plants from seeds. As a result, only one main root sample per clonal line was available for analysis, and increasing the number of replicates was not feasible. We have acknowledged this limitation and added a statement in lines 419–420 of the Discussion section to clarify this point.

Comment 6: Figure 1 (compound structure) is not cited in the main text. It is recommended that it be mentioned in the methods or results section or that Figure 1 be removed from the text.

Response: Thank you for pointing this out. We have added a reference to Figure 1 in lines 175–178 of the Results section to address this issue.

We look forward to hearing from you regarding our submission and would be happy to address any further questions or comments you may have.

Best regards.

Takahiro Tsusaka (TT), Miki Sakurai (MS)

---

## [Decision Letter · Decision Letter 1]

24 Jun 2025

Evaluation of genetic inheritance of the medicinal compound contents in Glycyrrhiza uralensis

PONE-D-25-10815R1

Dear Dr. Tsusaka,

We’re pleased to inform you that your manuscript has been judged scientifically suitable for publication and will be formally accepted for publication once it meets all outstanding technical requirements.

Kind regards,

Mehdi Rahimi, Ph.D.

Academic Editor

PLOS ONE

Additional Editor Comments (optional):

Reviewers' comments:

Reviewer's Responses to Questions

**Comments to the Author**

1. If the authors have adequately addressed your comments raised in a previous round of review and you feel that this manuscript is now acceptable for publication, you may indicate that here to bypass the “Comments to the Author” section, enter your conflict of interest statement in the “Confidential to Editor” section, and submit your "Accept" recommendation.

Reviewer #2: All comments have been addressed

2. Is the manuscript technically sound, and do the data support the conclusions?

Reviewer #2: Yes

3. Has the statistical analysis been performed appropriately and rigorously? 

Reviewer #2: Yes

4. Have the authors made all data underlying the findings in their manuscript fully available?

Reviewer #2: Yes

5. Is the manuscript presented in an intelligible fashion and written in standard English?

Reviewer #2: Yes

6. Review Comments to the Author

Reviewer #2: The authors have adequately addressed my comments raised in a previous round of review and I think that this manuscript can be accepted for publication.

7. PLOS authors have the option to publish the peer review history of their article (what does this mean? ). If published, this will include your full peer review and any attached files.

**Do you want your identity to be public for this peer review?** For information about this choice, including consent withdrawal, please see our Privacy Policy .

Reviewer #2: No

---

## [Editor Report · Acceptance letter]

PONE-D-25-10815R1

PLOS ONE

Dear Dr. Tsusaka,

I'm pleased to inform you that your manuscript has been deemed suitable for publication in PLOS ONE. Congratulations! Your manuscript is now being handed over to our production team.

Kind regards,

on behalf of

Associate Prof. Mehdi Rahimi

Academic Editor

PLOS ONE